# Until Death Do Us Part. The Influence of Own and Partner’s Socioeconomic Status on the Health of Spanish Middle-Aged Population

**DOI:** 10.3390/ijerph17134644

**Published:** 2020-06-28

**Authors:** Jordi Gumà, Jeroen Spijker

**Affiliations:** 1Department of Political and Social Sciences, Universitat Pompeu Fabra, 08005 Barcelona, Spain; 2Sociodemography Research Group (DEMOSOC), Universitat Pompeu Fabra, 08005 Barcelona, Spain; 3Centre for Demographic Studies, Universitat Autònoma de Barcelona, 08193 Bellaterra, Spain; jspijker@ced.uab.es

**Keywords:** partner, self-reported health, socio-economic status, sex differences, EU-SILC, Spain

## Abstract

Objectives: To explore whether the influence of a partner’s socioeconomic status (SES) on health has an additive or a combined effect with the ego’s SES. Methods: With data on 4533 middle-aged (30–59) different-sex couples from the 2012 Spanish sample of the European Union Statistics on Income and Living Conditions (EU-SILC) survey, we apply separate sex-specific logistic regression models to calculate predicted probabilities of having less than good self-perceived health according to individual and partner’s characteristics separately and combined. Results: Both approaches led to similar results: Having a partner with better SES reduces the probabilities of not having good health. However, the combined approach is more precise in disentangling SES effects. For instance, having a higher educated partner only benefits health among Spanish low-educated men, while men’s health is worse if they have a working spouse. Conversely, women’s health is positively influenced if at least one couple member is economically active. Conclusions: There are significant health differences between individuals according to their own and their partner’s SES in an apparently advantageous population group (i.e., individuals living with a partner). The combinative approach permits obtaining more precise couple-specific SES profiles.

## 1. Introduction

The protective health effect of living with a partner is well established and has been studied in detail from an individual approach. The literature generally tells us that the healthier profile of individuals who live with a partner is due to a range of factors that promote a healthier lifestyle and discourages individuals from pursuing risky behaviours. This includes a partner and social control through the creation and maintenance of social nets on which individuals can count on in case of economic and/or personal setbacks and material pathways of financial resources and economies of household scale, which may facilitate the purchases of better medical care, better diet, and safer surroundings [1]. A selection mechanism in the marriage market has also been proposed, whereby those with a good health profile have a higher probability of finding a partner and maintaining a relationship [2]. However, the majority of these studies considered living with a partner as a homogeneous situation for all couple members, after controlling for the individual’s characteristics, but without concomitantly taking into account the specific characteristics of the partner or the union [3].

To fill this knowledge gap, we therefore set out to test whether the health of those who live with a partner—a situation that pertains to 69% of the Spanish population aged 30–59 according to the European Union Statistics on Income and Living Conditions (EU-SILC) survey in 2012—not only depends on one’s own and the partner’s socioeconomic status (SES) characteristics, but also on the combined characteristics. The idea of combining information from both partners, instead of using individual information from each of the partners separately as in previous studies [4], is to see whether the frequently established positive association between individual SES—here considered to be educational attainment or employment status—and health remains the same after also considering the partner’s SES separately, or in combination with one’s own SES.

The majority of studies that tested the influence of a partner’s SES on individual health considered educational attainment as an indicator of SES due to its well-established, positive association with health and measured health outcomes according to mortality [4,5], subjective (i.e., self-rated) health [6,7,8,9], or risky behaviour [8,10]. All studies found that the inclusion of the partner’s education added meaningful information to better understand health inequalities, even after controlling for one’s own educational level: The higher the partner’s educational attainment, the lower the probability of dying prematurely, having a poor subjective health, or smoking. Based on the social causation hypothesis—which assumes that an individual’s education (or SES in general) affects material, behavioural, and psychosocial factors and that these in turn have effects on health—the partner’s educational attainment (or SES in general) must also have an additional effect on health, whereby the direction and intensity of the effect varies according to the kind and quantity of the contributed resources. Indeed, an individual’s financial situation, characteristics of the household they live in, and even opportunities within the labour market are ultimately defined by the features of both partners [8].

Previous studies also indicate that certain socioeconomic characteristics known to benefit health among individuals living in a couple differ by age, sex, or health outcome studied. For instance, according to Guallar-Castillón et al. [11], the contributing effects of not being the head of the family, having a lower educational level, and lifestyle-related variables on gender differences in a health-related quality of life (QoL) declined with age among the elderly in Spain (where women had substantially worse QoL than men). Similarly, an Israeli study [5] showed that, although the educational attainment of both spouses were significant predictors of one’s own overall mortality, the educational level of the spouse did not always have an effect when specific causes of death where analysed. For instance, the wife’s education had no effect on the spouse’s respiratory disease mortality or cancer mortality and the husband’s education had no effect on the wife’s respiratory and cardiovascular disease mortality. Skalická and Kunst [12] showed for Norway that, while the educational level of the wife is an important predictor of the husband’s mortality, the husband’s education is not a determinant of the wife’s mortality, but rather, the husband’s occupation and income are determinants. The authors suggested that the origin of this gender difference comes from the ‘gender-specific’ benefits that different socioeconomic sources appear to bring: While the husband’s contribution is based on the capacity to offer a certain degree of financial security (mediated through an occupational class), the wife’s education influences the husband’s health more than vice versa due to the wife’s greater involvement in domestic and care tasks (e.g., better knowledge about nutritional care or home organisation).

In addition, education is actually more relevant for women’s health because its influence on health is greater for persons with fewer alternative resources [13], especially through employment. Therefore, we assume that the partner’s SES characteristics —if at a higher level than the ego’s—can act as a resource to fill this gap. However, the opposite can also occur, i.e., having a low SES partner can aggravate the disadvantage that a low SES can have on health. In our opinion, this shows how important within-couple gender roles are in understanding socioeconomic differences in the health status of individuals living with a partner. Likewise, because the female educational expansion in Spain—as in other countries—has not yet been translated into gender equality in a labour-force participation, it underpins the necessity to, besides education, also consider the ego and partner employment status in a context of a gender-unbalanced labour force. One reason is because the partner employment status could be a complementary resource for many under- or non-employed women living as part of a couple.

Therefore, the aim of our study is to determine how, among Spanish co-residing couples, the combined SES of the individual (ego) and partner affect individual health status. We focus on the year 2012 when the recent economic recession was still in full force, making it more likely that partners’ characteristics influence individual health status. The last economic recession had a great impact on the Spanish population in terms of unemployment (reaching a maximum rate of 27% in early 2013 [14]) and loss of job security. This negative impact was greater among men in terms of job loss, forcing their partners to try to increase their participation in the workforce to compensate for the reduction in households’ income [15]. It is in this context that we consider that the characteristics of both partners gain in relevance.

We contribute to the existing research by considering the partner’s SES level both separately and combined with the ego’s characteristics. As resources can be better optimised when living with a partner, we assume that health differences between partnered individuals with the same SES could be explained by the SES of the partner. Specifically, we aim to answer the following research questions: Does combining individual and partner SES provide any additional explanation of socioeconomic differences in health beyond what is obtained by taking an additive approach? How does this vary by sex?

We restrict our analysis to Spain for two reasons: No study on the relationship between partner’s characteristics and individual’s health status has yet focused on a southern European country and, given its particular history, studying Spain provides the chance to test possible cohort-specific gender differences. Since the death of Franco, Spain experienced a process of rapid political and social change, including a diversification in family forms [16]. Educational expansion and the massive entry of women into the labour market also started later than in other Western countries [17], but it did so with higher intensity [18,19]. Accordingly, today’s middle-aged women are slightly more educated than their male counterparts, but Spain is still considered a country with an ‘in-complete gender transition’ because of persisting lower female labour force participation rates or lower male involvement in unpaid domestic work among dual-earner couples [20,21].

## 2. Materials and Methods

### 2.1. Data

The selected data source is the Spanish 2012 cross-sectional sample of EU-SILC that has information from private households on SES and health from all household members. Only native Spaniards aged 30–59 who live with a partner are analysed here. Younger adults were excluded as few have health problems or cohabit as well as the elderly to reduce selective mortality bias, which would complicate any comparison with younger ages. Excluded also were 60–64 year-olds (even if the younger partner was less than 60 years of age), as they are a very heterogeneous group in terms of employment status: Many retire before the statutory retirement age of 65 for economic reasons (e.g., imposed by their employer) or because of health issues (the average effective age at retirement in Spain was 62.2 for men and 63.1 for women for the period 2009–2014 [22]). Finally, 15 same-sex couples and 537 foreign and mixed couples were excluded to avoid possible bias due to the diversity in cultural and sociodemographic backgrounds, leaving 4591 Spanish mixed-sex couples, with the older partner aged 30–59. Of this subsample, 4533 (98.7%) had complete information on the selected variables.

### 2.2. Measures

Our dependent variable is self-rated health (SRH). SRH is known to capture health differences in a relatively homogenous middle-aged population in terms of objective health [23]. This information was obtained from the survey question: ‘How is your health in general?’. For the purpose of our study the possible answers ‘very good’, ‘good’, ‘fair’, ‘bad’, and ‘very bad’ were dichotomised into good health (‘good’ and ‘very good’ health) and less than good health (‘fair’, ‘bad’ and ‘very bad’ health) following the usual practice [24].

Our two main independent variables are educational attainment and employment status of both partners. The original seven education categories, based on the International Standard Classification of Education (ISCED), were aggregated into three: (1) Primary/lower secondary (ISCED 0–2); (2) upper secondary (ISCED 3–4); and (3) tertiary education (ISCED 5–6). The employment variable was derived from the self-defined current economic status question. The possible answers were collapsed into three categories: (1) Working (employee/self-employed working full/part-time); (2) unemployed; and (3) inactive (student/(early) retired/permanently disabled/fulfilling domestic tasks/other). Given the age- and sex-specificities of the inactive category, it was not further split up (i.e., most women in this category were homemakers (87%) and most men disabled (61%)).

All the models are controlled for ego’s age, marital status (married individuals are less likely to report a health-risk behaviour than non-married cohabiters [1]), the age of children at home as co-residing with children has an overall positive effect on health [25], but can also generate financial strain and perturb family-work balance [26], and partner’s SRH.

### 2.3. Statistical Analysis

Logistic regression models are used to identify which of the ego-partner combinations of educational attainment and employment status and categories of the co-variables are associated with less than good health among middle-aged individuals living with a partner. We use a sequential modelling strategy to assess the relevance of including combined individual-partner information. Men and women are analysed separately. Model 1 includes the ego’s educational level and employment status as well as the co-variables marital status, age of youngest child at home and health status partner, and the control variable, ego’s age. We subsequently test our additive model by including the partner’s information on educational attainment and their employment status (Model 2). The third and final model tests the combinative approach by including the combined (ego and partner’s) educational attainment and employment status variables. The main advantage of using this combined approach is that it permits identifying which combination of ego-partner SES characteristics benefits or harms health the most for someone who lives with a partner. All models were calculated using the software program STATA (version 14; StataCorp LLC, College Station, TX, USA). According to Mood [27], odds ratios reflect a certain degree of unobserved heterogeneity, thus preventing comparison across models and between (e.g., as in our case, sex-specific) samples. In line with the author, we therefore present the average marginal effects (AME), which for categorical variables correspond to the discrete change from the base level (henceforth reference category) in probabilities that the dependent variable is equal to 1.

Finally, as a robustness check, the analysis was repeated by performing an ordinal logistic regression by splitting up the two ‘less than good health’ categories into ‘fair’ and ‘poor’ health. Results showed that for both dependent variable categories virtually the same covariate category combinations were significant, as shown in Table 5 (these can be obtained from the corresponding author upon request).

## 3. Results

Table 1 presents the sex-specific distribution of the variables used in the analysis. Regarding the dependent variable, 85% of men and women aged 30–59 who lived as a couple were in good health. The absence of expected gender differences in health [28,29] can be explained by the fact that women in the sample were on average 2.2 years younger than their male partners (and age is negatively associated with health).

Turning to the independent variables, Table 2 shows the educational profile of both couple members separately (i.e., in the margin totals; also shown in Table 1) and according to the educational attainment of the partner. Noteworthy is that the studied age group has fewer low-educated women than men (40% versus 45%) but more men and women attained tertiary education or partnered with someone with a different educational level than older cohorts (results not shown).

If we consider the employment status profile of our middle-aged Spanish couples (Table 1 and Table 3), we can see how the higher educational expansion among women in Spain has not yet been translated into a situation of gender equality in the labour market. Among couples, men’s labour force participation rate exceeds women’s rate by 21.5 percentage points. Accordingly, five out of six partnered women who do work pertain to dual-earner couples. This unequal situation is mainly explained by the higher percentage of inactive women as the percentage of unemployment of men and women is similar (15.8% vs 19.6%).

Although not analysed in depth, another interesting result is that among 75.8% of couples both partners are in good or very good health (Table 4). Hence, the level of health homogamy is quite high and gender-symmetrical: Just 9.3% of men and 8.8% of women have better health than their partner. This is a plausible result given that the health status of women is compared with, on average, their slightly older male partners.

Moving to the multivariate analysis, Model 1 tests whether individual educational level and employment status have an effect on ego’s health. The other independent variables are also included in the model. The results for men (Table 5) show statistically significant differences in declaring less than good health between the reference category and the other response categories in both the ego’s educational attainment and employment status. In terms of AME, the result that stands out the most is that being inactive yields a 42% higher likelihood to declare less than good health compared to working. The results for women show a similar pattern, although with a lower magnitude. For instance, inactive women were only 7% more likely to not declare good health than those who were working.

In Model 2, we include the partner’s educational attainment and employment status. Both variables appear to barely affect the probability of not having good health. It is only significant for women whose partner has upper-secondary education and for men who live with inactive women. The individual-level effect of education and employment status remains virtually unchanged.

More interesting gender differences are observed in Model 3 that combines ego’s and partner’s educational attainment variables into one variable and likewise for employment status. Lower than tertiary-educated men have worse health (~10%), irrespective of their partner’s educational level, whereas for women the difference is significant when she declares to have upper-secondary education and her partner primary/lower secondary or primary/lower secondary and her partner lower than tertiary education. Turning to the combined ego-partner employment status variable, we observe how the reference category (both employed) shows the lowest probabilities of less than good health for both sexes. The only exception is employed egos with inactive partners, but only for men (−2%). There are also other gender differences. For men, to be employed appears to be the most advantageous situation regardless of their partner’s employment status. Conversely, unemployed men display higher probabilities of poor health when their partner is active (employed or unemployed). The same situation is not detrimental for women. For them, if just one couple member is employed (it does not matter who) their probability of less than good health is not significantly different than if both partners work. Additionally noteworthy is that not having good health is much more likely among inactive men than among inactive women, especially when the partner is unemployed (+52% vs. +11%) or works (+46% vs. +7%). Interestingly, for men the detrimental effect of being inactive is about 20% less severe when their partner is also inactive.

Finally, it should be noted that Model 3 explains more of the health differences in men (*r*^2^ = 0.18) than in women (*r*^2^ = 0.10).

## 4. Discussion

In this paper, we assessed the additive and combined effect of individual and partner educational attainment and employment status on self-perceived health among middle-aged Spanish couples in 2012 (close to the time when the last economic recession reached its peak), instead of the more common approach of analysing characteristics of both partners independently. As resources can be better optimised when living with a partner and education and employment status are known determinants of health at the individual level, we hypothesised that individuals with higher-educated or employed partners would have better health than their counterparts with equal or lower educational attainment or were not employed. If affirmative, this would imply that the known positive association between SES and health at the individual level could be mediated by the characteristics of the partner.

Results showed that the combined approach explained roughly the same variation in health as the additive models. However, the main advantage of opting for a combined approach comes from its capacity to identify which combined SES characteristics appear most beneficial or detrimental for the health of a partnered individual. For instance, and not surprisingly, a tertiary-educated ego partnered with someone who is also tertiary educated has better health than a low-educated individual with a likewise educated partner. In the case of low-educated men, however, another educational gradient can be discerned. For those who have a higher-educated partner, the detrimental effect of their own low education on health is reduced. Therefore, our results are only partially consistent with what Monden et al. [8] found for the Netherlands, and it is the only instance where having a higher educated partner appears to act as an additional resource for health. Conversely, Spanish men with upper secondary education have worse health when their partner has either the same or tertiary education than when she has a lower educational level. Among women with upper secondary education, we only observe higher probabilities of less than good health when their partner has primary/lower secondary education. Finally, for tertiary-educated egos, the effect of partner education on health is neutral.

However, in addition to Monden et al. [8]—who only focused on own and partner’s education—we also combined the employment status of both partners. This uncovered some interesting results. Although being the breadwinner of the household is most advantageous in terms of health for both Spanish men and women, one of the complementary situations (being unemployed when the partner works) is harmful for men’s health but appears of no importance for women. In other words, for Spanish women having at least one couple member economically active (it does not matter who) reduces the probability of having less than good health. Similar to education, the higher probability of being in less than good health in the case of unemployed men could be interpreted as a reflection or sign of frustration for those who assume or are expected to take on the role as a breadwinner [30], in addition to the known higher negative influence of unemployment on male mental health during the economic crisis [31]. Likewise, studies from the US indicate that middle-aged husbands who earn less than their wives have poorer physical health than similar husbands who are not secondary earners (for references, see Springer [32]). In Springer’s own attempt to disentangle the potential mechanisms responsible for this relationship, the author found that adverse health effects of income dependence especially persisted for high-earning men who believed in the male provider role and were possibly disappointed in not having a ‘housewife’. We also found considerable sex-differences in the magnitude of the probabilities of having less than good health among inactive individuals, although in this case we think that, besides the possible (self-perceived) lack of social approval, the origin of this inactivity may also play a crucial contribution: Most men declare being inactive due to health reasons, whereas women tend to be homemakers. Additionally, worthy of note is that the probability of having less than good health is lowest (albeit still substantially high) among inactive men when the partner is also inactive. While this suggests that men benefit more in terms of health if their partner also spends most of the time at home, a more detailed analysis showed that this was particularly the case for the older inactive men (50–59) who had left work for early retirement rather than for health reasons when their partner was also inactive (not shown here).

Our results must be also interpreted within the academic debate about gender difference in health perceptions. Despite the well-established sex differences in morbidity (women suffer more from non-acute disabling conditions whereas men present higher probabilities of having acute life-threatening conditions) and willingness to report health problems, the lack of difference in health status in our study population is nevertheless plausible given the fact women were compared with, on average, their slightly older male partners. Moreover, recent research has also casted doubts on the existence of gender-specific patterns in reporting either poor or good health [33], while in Spain sex differences in self-reported health were reduced once differences in socio-demographic characteristics, chronic conditions, and lifestyle behaviours between men and women were considered [34].

Lastly, our study has some weaknesses. EU-SILC does not contain information on health behaviour, including physical activity and the consumption of fruit and vegetables, smoking and alcohol, nor on family and employment trajectories that would allow possible selection mechanisms prior to partner co-residence or the effect of changes in working status on health to be studied.

## 5. Conclusions

Our study has shown that within an apparently advantageous population group in terms of health status—namely married or cohabiting couples—there are nevertheless significant health differences between individuals according to their own and their partner’s characteristics. Our results indicate that in a context of continuing educational expansion, the health of tertiary educated middle-aged women is less affected by the educational attainment of their partner than in the case of lower educated women. At the same time, having a higher educated partner especially benefits low-educated men. More concerning however, is with respect to employment, particularly among inactive men, who are much less likely to have good health irrespective of the employment status of their partner. We think that a selection effect likely plays a role here, as poor health could be the reason for not being active in the labour market, but this could not be confirmed with cross-sectional data. Another potential selection effect is partner’s health status. As a robustness check, we excluded this variable from our model and results showed that the detrimental health effect of having an inactive partner substantially increased among non-working women and inactive men. Moreover, having a non-healthy partner increases the chance of not being in good health, which we know can be due to shared living conditions [35], similar health-related behaviour [36], and possible health effects when having to care for an ill partner [37].

## Figures and Tables

**Table 1 ijerph-17-04644-t001:** Variables included in the analysis and the relative importance (%) of each category.

	Men	Women
**Dependent Variable**		
*Self-assessed health*		
Good health	85.3	84.8
Less than good health *	14.7	15.2
**Socioeconomic Variables**		
*Educational attainment*		
Primary or lower secondary	45.4	40.1
Upper secondary	22.4	23.7
Tertiary	32.3	36.3
*Employment status*		
Working	79.3	57.8
Unemployed	15.8	19.6
Inactive	4.9	22.6
**Other Covariables**		
*Marital status*		
Married	89.4	89.3
Not married	10.6	10.7
*Age of youngest child at home* ^#^		
Childless	23.0	21.5
Child(ren), youngest 0–2	9.9	9.9
Child(ren), youngest 3–15	42.5	43.3
Child(ren), youngest 16 +	24.6	25.3
*Health status partner*		
Good health	84.8	85.3
Less than good health	15.2	14.7
**Control Variable**		
*Age*		
<34	12.5	19.1
35–39	17.4	18.1
40–44	18.5	19.6
45–49	18.7	18.5
50–54	17.7	17.1
55–59	15.2	7.5
**Total Sample (N)**	**4533**	**4533**

Source: Spanish sample EU-SILC 2012. Note: Proportions are obtained from the weighted sample applying the survey’s personal cross-sectional weight, the reason why small differences are observed between men and women, and when they should be (and are in the unweighted sample) the same (e.g., with regard to marital status). * The combined categories of fair, poor, and very poor health. ^#^ Slightly larger sex differences in the percentages are observed for the variable ‘number and age of own children at home’. This is because in the survey children identify their parents, not the other way around. In the case of step-families, this means that not all step-parents are recognised as such.

**Table 2 ijerph-17-04644-t002:** Educational attainment and health status of individuals from co-residing couples aged 30–59. Spain 2012. Percentages.

		Female Partner			
		Primary or Lower Secondary	Upper Secondary	Tertiary	Total
	Primary or lower secondary	28.2	9.8	7.4	**45.4**
Male	Upper secondary	6.8	7.6	8.0	**22.4**
Partner	Tertiary	5.1	6.2	21.0	**32.3**
	**Total**	**40.1**	**23.7**	**36.3**	**100.0**

**Table 3 ijerph-17-04644-t003:** Employment status and health status of individuals from co-residing couples aged 30–59. Spain 2012. Percentages.

		Working	Unemployed	Inactive	Total
	Working	48.4	13.3	17.7	**79.3**
Male	Unemployed	7.5	5.2	3.1	**15.8**
Partner	Inactive	1.9	1.2	1.8	**4.9**
	**Total**	**57.8**	**19.6**	**22.6**	**100.0**

**Table 4 ijerph-17-04644-t004:** Self-assessed health status and health status of individuals from co-residing couples aged 30–59. Spain 2012. Percentages.

		Good or Very Good	Less than Good *	Total
Male	Good or very good	75.8	9.3	**85.1**
Partner	Less than good *	8.8	6.1	**14.9**
	**Total**	**84.6**	**15.4**	**100.0**

Source: EU-SILC 2012. Note. The marginal totals coincide with the sex-specific proportions in Table 1. * The combined categories of fair, poor, and very poor health.

**Table 5 ijerph-17-04644-t005:** Average marginal effect from the multivariate logistic regression analysis of less than good health of couples where the oldest partner is aged 30–59. Spain 2012.

		Men						Women					
		Model 1		Model 2		Model 3		Model 1		Model 2		Model 3	
		Coef	Sign	Coef	Sign	Coef	Sign	Coef	Sign	Coef	Sign	Coef	Sign
Educational attainment ego	Tertiary (Ref)	-		-				-		-			
	Upper Secondary	0.04	***	0.04	**			0.01		0.01			
	Primary/lower secondary	0.07	***	0.07	***			0.06	***	0.05	**		
Employment status ego	Working (Ref)	-		-				-		-			
	Unemployed	0.07	***	0.06	***			0.04	*	0.04	*		
	Inactive	0.42	***	0.42	***			0.08	***	0.08	***		
Educational attainment partner	Tertiary (Ref)			-						-			
	Upper secondary			0.00						0.04	*		
	Primary/lower secondary			0.01						0.02			
Employment status partner	Working (Ref)			-						-			
	Unemployed			0.00						0.02			
	Inactive			−0.04	***					−0.03			
Educational attainment combined	Both tertiary (Ref)					-						-	
	Ego tertiary-Partner upper secondary					0.03						0.04	
	Ego tertiary-Partner primary/lower secondary					0.04						0.00	
	Ego upper secondary-Partner tertiary					0.08	***					−0.01	
	Both upper secondary					0.07	*					−0.01	
	Ego upper secondary-Partner primary/lower secondary					0.03						0.07	**
	Ego primary/lower secondary-Partner tertiary					0.09	***					0.05	†
	Ego primary/lower secondary-Partner upper secondary					0.05	**					0.12	***
	Both partner primary/lower secondary					0.10	***					0.06	**
Employment status combined	Both employed (Ref)					-						-	
	Ego works-Partner unemployed					0.00						0.02	
	Ego works-Partner inactive					−0.03	*					−0.05	*
	Ego unemployed-Partner works					0.07	**					0.03	
	Both unemployed					0.05	†					0.05	*
	Ego unemployed-Partner inactive					0.02						0.03	
	Ego inactive-Partner Works					0.46	***					0.08	***
	Ego inactive-Partner unemployed					0.48	***					0.09	*
	Both inactive					0.25	***					0.07	†
	N	4533		4533		4533		4533		4533		4533	

Source: Spanish sample EU-SILC 2012. Significance of variable/category: *** p < 0.001; ** p < 0.001; * p < 0.05; † p < 0.01. Controlled for ego’s age, marital status, age of youngest child at home and health status of the partner. Data weighted by the survey’s personal cross-sectional weight.

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
