# Peer review of "Until Death Do Us Part. The Influence of Own and Partner’s Socioeconomic Status on the Health of Spanish Middle-Aged Population"

_ijerph, 2020, doi:10.3390/ijerph17134644_

Round 1
Reviewer 1 Report
INTRODUCTION:
- I would suggest breaking up the sentence below. It is much too long and dense.
"The literature generally tells us that the healthier profile of 29 individuals who live with a partner is due to a range of factors that promote a healthier lifestyle: 30 social and partner control, which discourages individuals from pursuing risky behaviours; creation 31 and maintenance of social nets on which individuals can count in case of economic and/or personal 32 setbacks; and economies of scale in the context of the household that optimise resources [1]."
- Line 53 - change to "the lower the probability of dying prematurely"
- Line 57 - change to "whereby the direction and intensity of the effect varies"
- Line 73-74 - change to "but rather, the husband’s occupation and income are determinants"
METHODS:
- Why are the data used 8 years old? Was there not any more recent data available?
- The inclusion of "fair" in the "bad" and "very bad" group" may dilute the magnitude. Why was "fair" not just dropped or treated at a third group? If it was used as a third group, then an ordinal logistic regression analysis could have been done in STATA (see - https://stats.idre.ucla.edu/stata/dae/ordered-logistic-regression/)
RESULTS:
- Table 3 is quite busy. I suggest creating two tables; one table for men and one for women.
DISCUSSION:
- There was not discussion on the variation of self-rated health between men and women, which may have influenced the initial response in the survey as well as the overall study results.
Author Response
Dear Reviewer,
We appreciate your positive evaluation and we consider it to be a significant motivation for our further work. We have carefully considered your comments and incorporated them into the revised manuscript hoping that the changes are in line with your ideas. The responses to your comments and suggestions can be found in the following text and all changes in the revised manuscript are highlighted using the track changes option in Word.
All reviews have been inspiring, and we see that this has greatly improved the quality of our paper. We would like to thank you for the recommendations and for the opportunity to revise our manuscript.
Kind regards,
Jordi Gumà
INTRODUCTION:
- I would suggest breaking up the sentence below. It is much too long and dense.
"The literature generally tells us that the healthier profile of 29 individuals who live with a partner is due to a range of factors that promote a healthier lifestyle: 30 social and partner control, which discourages individuals from pursuing risky behaviours; creation 31 and maintenance of social nets on which individuals can count in case of economic and/or personal 32 setbacks; and economies of scale in the context of the household that optimise resources [1]."
Reply: We’ve split the sentence up and made the sentences clearer:
“The literature generally tells us that the healthier profile of individuals who live with a partner is due to a range of factors that promote a healthier lifestyle and discourages individuals from pursuing risky behaviours. This includes partner and social control through the creation and maintenance of social nets on which individuals can count in case of economic and/or personal setbacks and material pathways of financial resources and economies of household scale, which may facilitate the purchases of better medical care, better diet, and safer surroundings.”
- Line 53 - change to "the lower the probability of dying prematurely"
Reply: Done.
- Line 57 - change to "whereby the direction and intensity of the effect varies"
Reply: Done.
- Line 73-74 - change to "but rather, the husband’s occupation and income are determinants"
Reply: Done.
METHODS:
- Why are the data used 8 years old? Was there not any more recent data available?
Reply: The reviewer is right to point that we did not explain at all why we decided to analyze the data of 2012. The main reason is because this was around the worst moment of the recent economic recession in Spain, a context in which we think that the characteristics of both partners would be important for understanding health differentials. We have now mentioned this in the manuscript.
- The inclusion of "fair" in the "bad" and "very bad" group" may dilute the magnitude. Why was "fair" not just dropped or treated at a third group? If it was used as a third group, then an ordinal logistic regression analysis could have been done in STATA (see - https://stats.idre.ucla.edu/stata/dae/ordered-logistic-regression/)
Reply: The reason why the “less than good” category was created was because this category contains just 15% of the male and female samples, as the fast majority of middle-aged individuals tend to have a good health profile. If we would have split this category up we would have had categories with too few cases to produce reliable results, especially once they would be crossed with covariates. Indeed, we followed the common practice in dichotomizing the information about self-perceived health as stated by Simone Croezen et al. (2016) in their paper “Self-perceived health in older Europeans: Does the choice of survey matter?”.
Just to make sure the results are robust, we calculated the models again following the reviewer’s suggestion using ordinal logistic regression. The following table confirms that both approaches lead to the same results in terms of the magnitude and statistical significance for the coefficients. We can observe that most categories that are significant in the ordinal regression model are also significant in the original model. In the majority of the cases, the Average Marginal Effects (AMEs) from the logistic regression add up to the AMEs for the categories “fair” and “poor” from the ordinal regression models both for men and women. A note of this robustness check has been provided in the paper.
Table 1. Average Marginal Effect from multivariate ordinal logistic regression analysis of fair and poor health of couples where the oldest partner is aged 30-59. Spain 2012.
Please see the attached table
RESULTS:
- Table 3 is quite busy. I suggest creating two tables; one table for men and one for women.
Reply: Done
DISCUSSION:
- There was not discussion on the variation of self-rated health between men and women, which may have influenced the initial response in the survey as well as the overall study results.
Good point. We did mention in the results section that the lack of gender differences in self-perceived health was “a plausible result given that the health status of women is compared with, on average, their slightly older male partners”. But you are right to suggest that men and women may also perceive their health differently. We have added this point to the conclusions of the manuscript, stating that previous research found that sex-differences in self-reported health in Spain is reduced once differences in socio-demographic characteristics, chronic conditions, and lifestyle behaviours between men and women were considered.

Reviewer 2 Report
This is an interesting study on the potentially more than additive effect of the partner's educational attainment or employment status on subjective health of the partner among 30 - 59 year old different sex couples in Spain. However, according to my evaluation the study has one serious shortcoming that is reported by the authors but in my opinion somewhat underrated from the perspective of significance. The data the analysis is based on, i.e. the European Union Statistics on Income and Living Conditions survey (EU-SILC) does not include information on central health behaviour which might be an important confounder with regard to the main findings. Moreover, another in my opinion rather important limitation is that poor health could be the reason for not being active on the labour market, but this could not be confirmed with cross-sectional data but is only mentioned in the Conclusions. However, there the authors report about a robustness check in which this variable was excluded from the statistical multi-variable model and results showed that the detrimental health effect of having an inactive partner substantially increased among non-working women and inactive men.
The study is well written but the authors cannot do anything to the limitations posed by their data and hence, I leave the final decision of publication to the Editorial Office.
Author Response
Dear Reviewer,
We appreciate your positive evaluation and we consider it to be a significant motivation for our further work. We have carefully considered your comments and incorporated them into the revised manuscript hoping that the changes are in line with your ideas. The responses to your comments and suggestions can be found in the following text and all changes in the revised manuscript are highlighted using the track changes option in Word.
All reviews have been inspiring, and we see that this has greatly improved the quality of our paper. We would like to thank you for the recommendations and for the opportunity to revise our manuscript.
Kind regards,
Jordi Gumà
This is an interesting study on the potentially more than additive effect of the partner's educational attainment or employment status on subjective health of the partner among 30 - 59 year old different sex couples in Spain. However, according to my evaluation the study has one serious shortcoming that is reported by the authors but in my opinion somewhat underrated from the perspective of significance. The data the analysis is based on, i.e. the European Union Statistics on Income and Living Conditions survey (EU-SILC) does not include information on central health behaviour which might be an important confounder with regard to the main findings.
Reply: We agree with the reviewer’s observation that unfortunately EU-SILC does not contain information on health behaviours. We have now mentioned this as a limitation.
Moreover, another in my opinion rather important limitation is that poor health could be the reason for not being active on the labour market, but this could not be confirmed with cross-sectional data but is only mentioned in the Conclusions.
Reply: This was mentioned in the conclusions as a limitation to the study. EU-SILC does not ask the respondent whether poor health is a reason for not being active on the labour market, so we could not confirm causality. We have now mentioned this as a limitation. However, health differences between employment statuses are very conclusive.
However, there the authors report about a robustness check in which this variable was excluded from the statistical multi-variable model and results showed that the detrimental health effect of having an inactive partner substantially increased among non-working women and inactive men.
Yes, with the robustness check where we took out the variable “partner health” to check if partner health has a potential selection effect on respondent’s health according to employment status, we saw that the detrimental health effect of having an inactive partner substantially increased among non-working women and inactive men. This supports earlier findings that have shown that this can be due to shared living conditions, similar health-related behaviour and possible health effects when having to care for an ill partner.
Round 2
Reviewer 1 Report
The edits and changes have improved the manuscript. Congratulations!